# Modulating Tumor Cell Functions by Tunable Nanopatterned Ligand Presentation

**DOI:** 10.3390/nano10020212

**Published:** 2020-01-26

**Authors:** Katharina Amschler, Michael P. Schön

**Affiliations:** Department of Dermatology, Venereology and Allergology, University Medical Center Göttingen, Robert Koch Str. 40, 37075 Göttingen, Germany

**Keywords:** biophysical cues, extracellular matrix, nanostructured ligand presentation, tumor progression, biophysical toxicity

## Abstract

Cancer comprises a large group of complex diseases which arise from the misrouted interplay of mutated cells with other cells and the extracellular matrix. The extracellular matrix is a highly dynamic structure providing biochemical and biophysical cues that regulate tumor cell behavior. While the relevance of biochemical signals has been appreciated, the complex input of biophysical properties like the variation of ligand density and distribution is a relatively new field in cancer research. Nanotechnology has become a very promising tool to mimic the physiological dimension of biophysical signals and their positive (i.e., growth-promoting) and negative (i.e., anti-tumoral or cytotoxic) effects on cellular functions. Here, we review tumor-associated cellular functions such as proliferation, epithelial-mesenchymal transition (EMT), invasion, and phenotype switch that are regulated by biophysical parameters such as ligand density or substrate elasticity. We also address the question of how such factors exert inhibitory or even toxic effects upon tumor cells. We describe three principles of nanostructured model systems based on block copolymer nanolithography, electron beam lithography, and DNA origami that have contributed to our understanding of how biophysical signals direct cancer cell fate.

## 1. Setting the Stage: Environmental Signals Modulate Cellular Functions

Cells use membrane receptors like integrins and cadherins to communicate with each other as well as with their environment (Figure 1). These ligand interactions are crucial for maintaining cell viability and structural tissue integrity [1,2]. On the molecular level, membrane ligands tend to cluster upon activation. Intracellular proteins bind to the inner domain of clustered ligands, thus leading to focal adhesion formation and the activation and re-organization of the cytoskeleton. The signals generated by the adhesion-related molecular machinery are linked with classical signaling pathways controlling cell growth and differentiation [1,3]. In a very simplified way, this is how biophysical signals are integrated to create a situational adaptation of cellular behavior and how fundamental cellular responses to environmental cues are processed.

An instructive example for the relevance of serially ordered and specific ligation of ligands that induces distinct cellular responses is leukocyte extravasation, a process that occurs in distinct steps of interactions with the endothelial lining, i.e., tethering, rolling, firm adhesion, and, eventually, transmigration [4,5,6]. This process is responsible for the controlled attraction of immune cells to sites of inflammation. It illustrates how ligand interactions direct immune cells from the blood stream to the target sites. Likewise, tumor cells utilize such cellular ligand interactions for spreading, invasion, extravasation and migration through tissues [7,8,9]. For example, melanoma cells (metastasized melanoma is one of the deadliest tumor entities) exhibit an array of upregulated and functionally activated adhesions receptors allowing them to metastasize to many different tissues [10,11]. Despite recent improvements in melanoma treatment through targeted therapies with small molecules [12] and immune-checkpoint blockade [13], the tumor is still fatal to most patients with advanced disease [14,15]. In this light, it is crucial to know as much as possible about the biomechanical and pathophysiological factors that influence the progression of melanoma (and other tumors). The modulation of the micromilieu of tumors may have a considerable influence on the survival of tumor cells and the effect of cytotoxic therapeutics. Indeed, the impact of distinct features of the ECM (extracellular matrix) on cell growth, survival, migration and differentiation has become increasingly clear in recent years [16]. The tumor–stroma interaction involves a complex interplay between tumor cells, fibroblasts, endothelial cells, immune cells, and the extracellular matrix. Melanoma cells have been shown to modify the tumor environment in a paracrine fashion and thereby create permissive situations for cell invasion and tumor progression [8]. These processes include the modification of ECM features on the nanoscopic scale, such as the composition and gradual availability of integrin ligands that have been shown to control tumor cell motility, apoptosis, and migration [17,18]. Consequently, blocking agents have been developed to interrupt the interaction of for an example melanoma cells with the extracellular matrix [19,20,21,22]. Unfortunately, the natural environment of a tumor is very complex and difficult to study, so standardized and tunable models are needed to specifically study the exact function of progression-related adhesion molecules such as integrins and cadherins [19,20,21,22].

Defined nano-structured matrices have made a significant contribution to uncovering some surprising functions of adhesion receptors, which can partly provide a biophysical basis or explanation for the effect or ineffectiveness of therapeutics directed against them [23,24,25]. Indeed, the spatiotemporal variation and distribution of ligands is highly relevant for distinct tumor cell functions [26,27,28]. Accordingly, there is a considerable need to mimic these biophysical parameters and investigate their respective influence on ligand interactions with respect to tumor progression on the one hand and inhibiting or cytotoxic activities on the other.

In the following, we give a brief overview of general biophysical features of the extracellular matrix (ECM), and we highlight the relevance of defined ligands and mechanotransducers in physiologically relevant settings. We then discuss several nanostructured models which enable the controlled site-directed presentation of ligands. Finally, we describe tumor progression-related functions that have been refined through the precise modulation of ligand density on the nanoscale. We will also discuss the utility of such systems to study either progression-promoting or -inhibiting (cytotoxic) mechanisms.

## 2. Biophysical Properties of the Extracellular Matrix—More Than Just a Scaffold

The ECM conveys and enables the exertion of biophysical functions by cells, for example the use of traction and propulsive forces or motility as well as survival or death signals through the presentation of certain ligands. Tissue rigidity, ligand topography, density and spatial distribution play decisive roles [29,30,31,32].

The stiffness and elasticity of the ECM is determined by the composition and cross-linking of collagen and other structural proteins. Different tissues display characteristic stiffness values usually specified by the elasticity (or Young’s) modulus (measured in Pa) [33]. All organ-resident and immigrating cells are exposed to the respective isometric forces generated locally at the nanoscale level by intercellular or cell–ECM contacts [34]. Cell are usually adapted to their specific tissue of origin. With regard to tumors, tissue stiffness is only one parameter that influences progression (melanomas, for example, metastasize into both very hard bones and soft brain tissue). However, tumor derived factors may enhance the stiffness of the ECM, as has been shown for both breast cancer and melanoma [35,36]. In pancreatic carcinoma, this has been viewed as a defense mechanism against immune cells [37]. Therefore, the response of tumor cells and immune cells to ECM stiffness variations is relevant for the understanding of tumor progression.

The nanoscale topography and structure of extracellular matrix components is difficult to decipher [29]. Usually, *ex vivo* tissue samples are used to analyze the architecture (this approach, however, is hampered by potential artifacts occurring during fixation). Both the size and the three-dimensional spatial arrangement of extracellular structures determine the presentation of ligands to cells [29,38]. Indeed, the topography of ligands has been shown to decisively control cellular functions associated with tumor progression like migration, growth or differentiation, and, as a consequence, their potential response to cytotoxic signals [31,39]. Of particular importance is the local density of ligands. Such biophysical features are relevant for many cellular interactions and are the focus of our review.

## 3. YAP and TAZ—Master Regulators of Mechanosensing

Understanding how cells sense biophysical cues of their environment and transduce them into gene regulation is a prerequisite for the modulation of complex functions including motility, proliferation or death. Indeed, environmental characteristics like the spatial distribution of ligands regulates the clustering of cellular receptors which in turn leads to the activation of the cytoskeleton [2]. This notion underlines the relevance of specific ligand presentation for mechanotransduction as for example intracellular proteins like talin are thought to arrange themselves according to the spatial positioning of integrins [40,41,42]. In addition, the transcription factors YAP (yes-associated protein) and TAZ (WW domain containing transcription regulator-1) are key to mechanosensing in physiological situations but also in pathophysiological processes like cancer progression [43,44,45]. Both YAP and TAZ are part of the Hippo pathway which is relevant for organ size control through the regulation of proliferation and apoptosis [46]. However, during mechanosensing YAP and TAZ seem to act independently of the Hippo pathway [44,47]. Activation of YAP/TAZ is primarily regulated by their subcellular distribution in the cytosol (inactivated) versus transfer into the nucleus (activated). Several biophysical parameters such as tissue stiffness, shear stress or spatial distribution of ligands have been shown to facilitate the transfer of YAP into the nucleus where it binds to other proteins altering gene expression. Interestingly, f-actin is indispensable for the activation of YAP/TAZ although it is still unclear how its assembly to stress fibers leads to YAP/TAZ activation [43,44].

YAP and TAZ are upregulated in tumor cells and have been implicated in stem cell attributes, proliferation, chemoresistance and metastasis [48,49]. In malignant melanoma, they convey resistance to BRAF/MEK inhibition [50]. Intriguingly, they also increase mechanosensitivity or even independence of external growth stimuli [51]. In fibroblasts within the tumor stroma, YAP/TAZ activation leads to increased collagen deposition and, consecutively, to enhanced organ stiffness. The latter in turn activates YAP and TAZ in tumor cells [48,50]. Therefore, YAP and TAZ play a fundamental role in choreographing tumor-stromal interactions, and defined modulations of the extracellular matrix that influence the function of YAP and TAZ could therefore improve the effect of antitumor cytotoxic therapies [49]. Therefore, it is particularly important to use tunable and standardized model systems of defined nanostructured ligand presentation for the investigation of important cell functions.

## 4. Nanoscale Ligand Control in Experimental Models

The true two- and three-dimensional ligand distribution as well as the actual presentation of ligand recognition motifs in biological matrices are difficult to determine. However, recent advances in imaging techniques have allowed to estimate some in vivo ligand densities. Although there are of course differences between different receptor-ligand pairs, most of the interactions investigated so far seem to take place at lateral distances between 20 and 200 nm [24,52,53,54]. Therefore, model systems with ligand presentation in this order of magnitude will be outlined in the following.

Two general methods are mainly used for the generation of nanostructured matrices: The first are the so called “top-down” methods which produce nanostructures by comminuting larger materials through lithographic tools such as electron-beam lithography. On the other hand, the so called “bottom-up” techniques use the self-assembly of molecules to create the desired nanoscale structures. The latter include the DNA-origami method and block copolymer nanolithography [55]. Both principles can also be combined for the fabrication of more sophisticated nanopatterns [56].

Most nanomodels for studying and manipulating cell functions use gold (Au) nanoparticles as an anchor because Au binding to thiol groups provides for a covalent ligation required for the immobilization of different molecules. In addition, Au nanoparticles allow the site-directed presentation of proteins through a nickel-his-tag complex. This is an elegant and versatile method to mimic the site-directed display of biomolecules in natural matrices such as plasma membranes or extracellular matrices [53,57]. All systems require an inert background preventing unspecific interactions. The proper orientation of transmembrane proteins, however, is more challenging [57]. Experimental systems become even more complex when they are aimed at: (i) the presentation of more than one ligand (monovalent vs. multivalent), (ii) ligand presentation in a particular distribution such as clustered ligands, and/or (iii) variations of substrate elasticity/stiffness (e.g., different glass or gel matrices, depending on the desired stiffness). Until now, it is not possible to adjust all these parameters simultaneously.

Block copolymer (micellar) nanolithography (BCML, Figure 2a) is an appropriate method to deposit metal particles on substrates in a quasi-hexagonal order, thus allowing the local and global control of nanoparticles. The basis of this technique is the self-assembly of block copolymers to micelles in a non-polar solution. The hydrophilic center of the micelles can be loaded with metallic nanoparticles and the distance of the nanoparticles can be adjusted by the length of the polymers. Spin-coating or dip-coating (the velocity of dipping controls the distance of the nanoparticles) mounts the nanoparticles onto (glass) substrates. Subsequent cold plasma treatment removes the polymers. The particles thus immobilized in a structured pattern can then be used as anchor molecules for biofunctionalization with a variety of molecules. In principle, it is possible to deposit different metallic nanoparticles. An inert surface with Pll-g-PEG (poly L-lysine-grafted-polyethylenglycol) prevents unspecific protein adsorbance or cell binding in-between the gold nanoparticles. The Pll-g-PEG itself can also be spiked with additional ligands or ligands can be activated by click chemistry, thus creating bi- or multifunctional matrices [53,58,59,60,61].

The second method that is widely used to create nanopatterns to address biological questions is electron beam lithography (EBL, Figure 2b). This top-down technique uses a focused electron beam to draw nanoscopic patterns on a surface covered with an electron-sensitive layer [62,63,64]. The latter is called a resist. The solubility of the resist is altered by the electron beam, thus allowing selective solubilizing of either the exposed or non-exposed areas. This creates nanoscopic structures in the resist with sub-10 nm resolution. Ion beam lithography may yield even higher resolution. The modified resist is then covered with a metal filling the nano-holes. After lifting off the resist either the exposed or non-exposed regions remain on the substrate in the previously drawn pattern. Again, gold is usually used for biological experiments. The so produced Au nanopatterns can be further modified by pegylation using Pll-g-PEG [65] or lipid bilayers spiked with a second ligand [66]. A considerable disadvantage of EBL is its production time as for example 1 cm^2^ large nanopatterns may take 4 days of electron beam writing [67]. Even though progress has been made towards high-speed production [68], it is not an appropriate method for mass production.

The creation of nanopatterns with DNA origami (Figure 2c) is a bottom-up method using the flexible foldability of single-strand DNA. Multiple defined short pieces of corresponding DNA strands bend it to form a well-defined molecule on the nanoscale. Computer-assisted prediction of DNA double-helix folding provides access to multiple variable 2D and 3D nanostructures called DNA origami [69,70,71]. Various functional groups can be incorporated to enable reactions with surfaces (for example covalent immobilization on a substrate) or with cells. In this way, DNA origami have been functionalized with peptide recognition motifs with a well-defined spatial distance of 60 nm and have been patterned on substrates by covalent ligation to study cell adhesion processes on patterned matrices [66].

## 5. Modulation of Tumor Cell Functions by Nanostructured Ligands

Biophysical traits of the microenvironment are crucial for tumor formation [16,72]. They can be influenced by nanoscopic ligand variation. For example, tumor cells can stimulate their own growth through autocrine mediators entrapped within the extracellular matrix [72]. Ligands like Epidermal Growth Factor (EGF) bind to cellular receptors which often oligomerize and activate signaling cascades. DNA origami in combination with electron beam lithography has demonstrated the relevance of the EGF ligand architecture and nanoscale distribution for the elicitation of cellular responses [55]. In addition, DNA origami-based multivalent nanomodels unraveled cooperative functions of EGF and integrin ligands immobilized at distances of 60 nm. These insights highlight the importance of cooperative ligand functions for central tumor cell functions such as proliferation and spreading [73].

Epithelial tumor cells reduce their cell-cell adhesion, lose their cellular polarity and acquire a migratory and invasive phenotype in a process called epithelial-to-mesenchymal transition (EMT) [74]. Higher density of certain ligands within the ECM may facilitate EMT and weaken intercellular contacts between tumor cells [26]. This effect can be modeled on RGD (arginine-glycine-aspartate)-coated surfaces of different densities showing that RGD rich substrates retained cells in the epithelial phenotype while reduced RGD density promoted EMT [75].

Tumor cell invasion is promoted by invadopodia formation, actin-rich membrane protrusions with longitudinal cytoskeletal structures and proteolytic capacities to transform (degrade) the ECM. Interestingly, invadopodia formation can be initiated by large distances between certain ligands (i.e., low ligand density), as shown by fibronectin-coated gradient nanopatterns created by electron beam lithography [62]. Similar fibronectin-coated gradient nanopatterns demonstrated that breast cancer cells but not normal mammary cells can adapt flexibly to different ligand densities [63]. Marked cellular flexibility on variable ligand densities was also detected in melanoma cells exposed to Au nanopatterns functionalized with N- or E-cadherin. Indeed, melanoma cells expressing the corresponding cadherin showed similar cytoskeletal organization and spreading on ligand densities ranging over one order of magnitude [76] (Figure 3). In addition, the use of this nanotechnological platform revealed an unexpected heterophilic interaction of E-cadherin-only expressing melanoma cells with immobilized N-cadherin at high densities. Such unexpected findings bear implications for tumor cell behavior in vivo and would not have been possible without the use of nanotechnology.

While nanotechnological approaches uncovered the capacity of tumor cells to flexibly adapt to different matrix conditions, this trait apparently depends on the specific ligands studied. When BCML-derived Au nanopatterns with tunable densities ranging from ca. 70 to 1145 ligand sites/µm^2^ (corresponding lateral distances of 120–30 nm) were functionalized with cyclic RGD (which binds to the α_V_β_3_ integrin on melanoma cells), an intermediate ligand density of about 365 ligands/µm^2^ yielded maximum cell spreading as well as activation of the cytoskeleton. Higher or lower densities, respectively, led to reduced spreading [77] (Figure 3). When Vascular Cell Adhesion Molecule (VCAM)-1 (an endothelial ligand of the melanoma cell-expressed α_4_β_1_ integrin) was presented on the same nanostructured matrix at densities ranging from 70 to 1145 ligands/µm^2^ (corresponding lateral distances of 120–30 nm), a density-dependent inhibitory effect on cell spreading and cytoskeletal activation was observed. This unexpected “paradoxical” effect could be abrogated by the additional presentation of RGD in a bifunctional matrix but only when the VCAM-1 density was reduced to 120/µm^2^ (lateral distances of 90 nm) [78]. Both observations, non-linear behavior of tumor cells and “paradoxical” inhibition of cell spreading, respectively, were made possible by nanotechnology. They shed light on hitherto difficult-to-study tumor cell functions and provide a potential explanation for the “paradoxical” enhancement of melanoma metastasis by the specific RGD inhibitor, cilengitide, at suboptimal doses [79]. With murine melanoma cells, the inhibition of spreading by VCAM-1 observed with nanostructured matrices also corresponded to the function of these melanoma cells observed in vivo [78]. The differential influences of adhesion ligands on melanoma cell behavior are exemplified in Figure 3.

In general, biophysical parameters such as ligand density as well as ligand specificity determine important progression-related functions of tumor cells. On the one hand, tumor cells can adapt flexibly to different ligand site densities [63,76]. On the other hand, ligand density variations can induce phenotypic switches resembling EMT [80]. In a context-dependent manner, certain ligand constellations seem to provide either permissive or inhibiting signals to tumor cells. Misdirected interactions of tumor cells with the ECM are relevant for tumor progression. Accordingly, cancer can be defined as “reciprocal interactions among cells” that went out of control [81]. Nanotechnology has helped to identify some of these biophysical cues that direct cancer cell fate, and it can thus help to identify environmental conditions that render tumor cells susceptible or resistant to cytotoxic therapies. Some examples of such applications [55,62,63,73,76,77,78] are summarized in Table 1.

## 6. Limitations of Current Nanomodel Systems

Even though a number of tumor-relevant functions have been unraveled using nanomodel systems, the current methods have some limitations. First, the model systems described usually work with two dimensions which of course do not reflect the complex 3D situation in vivo. Second, a major issue lies in the uncertain stability (over time) of the model-systems during the experiments. Therefore, the read-out is usually focused on short-term cellular modifications while long-term analysis (several days and weeks) including proteome modification are difficult to analyze. Toward that end, optical nanoscopy has improved the illustration of subcellular organelles and the cytoskeleton [82,83]. Finally, it is a challenge to transfer the experimental data of nanomodels to the in vivo situation as the true nanoscopic in vivo ligand density is still difficult to determine.

## 7. Biophysical Cues Modulate Cell Death and Survival

Biophysical parameters can either enhance or diminish cell viability and vulnerability, a notion that is interesting from a therapeutic point of view. The maximum form of biophysically mediated cell death is called anoikis [84]. In general, cells rely on signals from their environment conveyed by ligand interactions. In particular, the availability of integrin ligands promotes cell survival and proliferation [85]. If cells lose their physical contact with ligands they die—a critical mechanism that prevents cell growth and dissemination outside their tissue of origin [84]. Many tumor cells are able to resist this form of controlled cell death and adapt to a reduced availability of ligands [63,86,87,88]. However, integrin ligands might also exert subtle inhibitory effects controlled by ligand density variation as has been exemplified by the inhibition of melanoma cell spreading induced by VCAM-1 presented at high densities [78]. In fact, integrins have been shown to actively induce cell death when appropriate ligands are lacking. Interestingly, this form of cell signaling is also directed by physical cues which are accessible by nanotechnological approaches [89,90]. Thus, the decision to live or to die is not only mediated by cytotoxic ligands such as FasL interacting with its receptor Fas—a classical pathway of apoptosis [91]. Accordingly, cells integrate multiple biochemical and biophysical cues to generate a response adapted to the distinct challenges. As a consequence of the considerations presented in this review, it is reasonable to assume that biophysical signals antagonize or potentiate classical cell death signals like Fas/FasL, TNF mediated signaling pathways, or antitumoral cytotoxic therapies [92,93].

## Figures and Tables

**Figure 1 nanomaterials-10-00212-f001:**
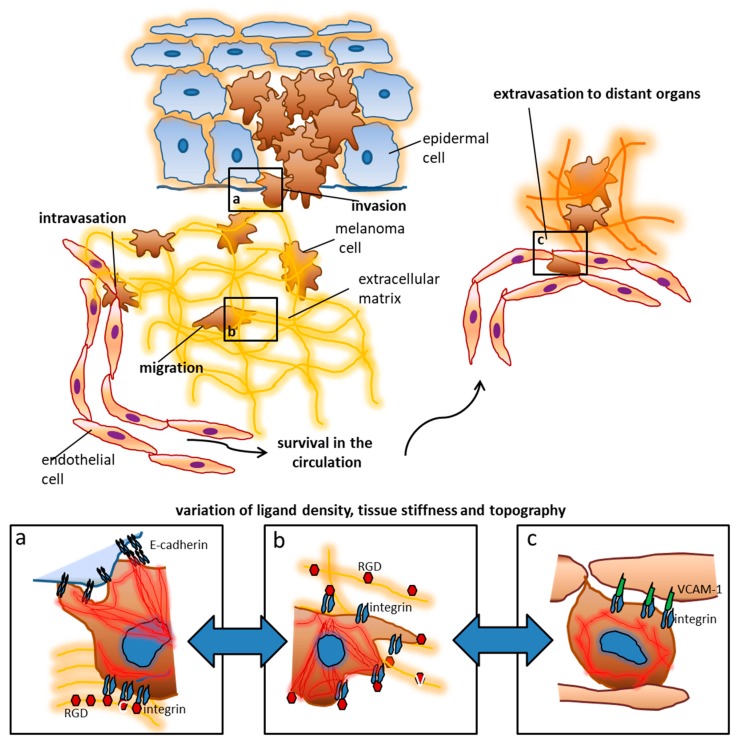
The extracellular matrix and stromal cells influence the behavior and fate of tumor cells. Using melanoma as an example, early tumor stages are controlled by contact with keratinocytes, which is mediated by cadherins and other surface receptors. During invasion, melanoma cells reach the extracellular matrix where they encounter additional ligands like RGD (arginine, glycine, aspartate) motifs in different arrangements. These ligands enable migration towards blood vessels and lymphatics. Upon intravasation tumor cells become exposed to new biophysical influences including shear and tensile forces by the blood flow. Adhesion ligands in distant organs like Vascular Cell Adhesion Molecule (VCAM)-1 lead to adhesion and tumor cell arrest which is a prerequisite for extravasation and metastasis formation. The colonization of distant organs requires the tumor cells to deal with very different ECM (extracellular matrix) conditions which include tissue stiffness, ligand density, and topography. The panels a–c depict some examples of tumor cell interactions which can be addressed by nanotechnological approaches: (**a**) when tumor cells leave their original tissue, cadherin interactions are presumably involved. Changes in the cadherin repertoire as well as the density and topographical arrangement of the ligand presentation play an important role according to our current knowledge. Upon contact with high densities of RGD, melanoma cell spreading is inhibited, while E-cadherin (largely irrespective of its density) enhances spreading. (**b**) In the ECM of the connective tissue, integrins and gradients of their ligands are particularly responsible for motility, morphogenesis, and spread of tumor cells. These functions are for example clearly controlled by RGD residues. Melanoma cells entrapped in an ECM with reduced RGD ligands show maximum spreading possibly enhancing migration. (**c**) It is assumed that VCAM-1 density plays an important role in the extravasation of melanoma cells at sites of metastasis formation, whereby higher VCAM-1 densities can inhibit the spreading of tumor cells.

**Figure 2 nanomaterials-10-00212-f002:**
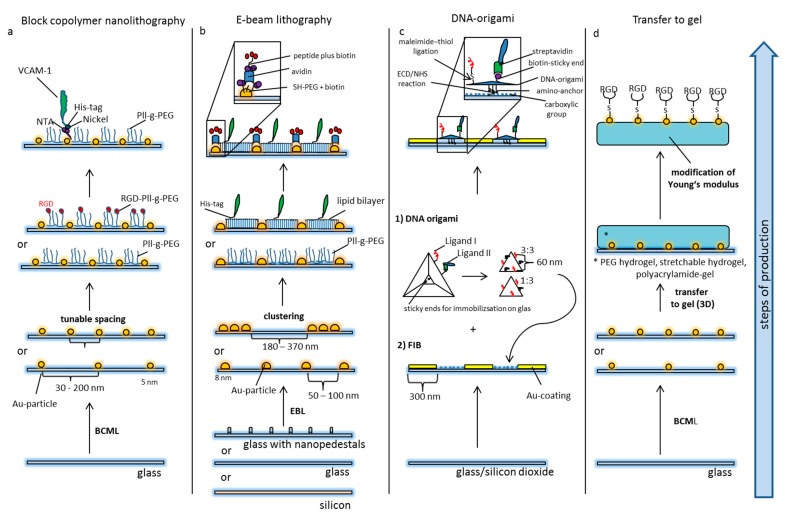
Principles of nanostructured models of cellular and extracellular matrices. (**a**) Block-copolymer nanolithography (BCML): Glass slides are covered with 5 nm gold particles by BCML. The advantages of this method are the high production rate and the flexible tunable densities of Au nanoparticles. The area in-between the nanodots is passivated by pegylation preventing unspecific protein adsorption. The Pll-g-PEG layer can also be spiked with additional ligands, or ligands can be unmasked via click chemistry. The Au-particles serve as anchor points for the site-directed display of ligands via creating an NTA–Nickel–His–tag complex. (**b**) Electron beam lithography: Glass slides are nanopatterned by electron beam lithography. Based on different production processes, modifications like nanopedestals (for the axial presentation of ligands) or alternative materials like silicon have been used. The general design of the patterns (distribution of particles) can vary. Pll-g-PEG in-between the particles prevents unspecific binding, or a second ligand can be presented using artificial lipid bilayers. The Au-particles can be functionalized by thiol–PEG carrying biotin moieties. The latter can ligate avidin for immobilization of biotin-tagged peptides or proteins. (**c**) Focused ion beaming in combination with DNA origami: For the immobilization of DNA origami, glass substrates containing silicon dioxide are patterned with an Au-surface using focused ion beaming (FIB). The silicon in-between is further prepared to present carboxylic groups for the covalent ligation of the amino anchors of the DNA-origami (EDC-NHS reaction). The DNA origami itself is produced by the self-assembly of single DNA strands which are forced into a certain configuration by short congruent DNA strands. The distance of functional groups can be pre-determined on the nanoscale, and the amount and relation of ligands can be adjusted. Different ligands can be functionalized by distinct forms of peptide immobilization, e.g., maleimide-thiol ligation on the one hand and biotin-streptavidin ligation on the other. (**d**) Transfer of nanopatterns to gel: Nanopatterns can be transferred to different gels. Especially nanopatterns produced by BCML have been used for this purpose.

**Figure 3 nanomaterials-10-00212-f003:**
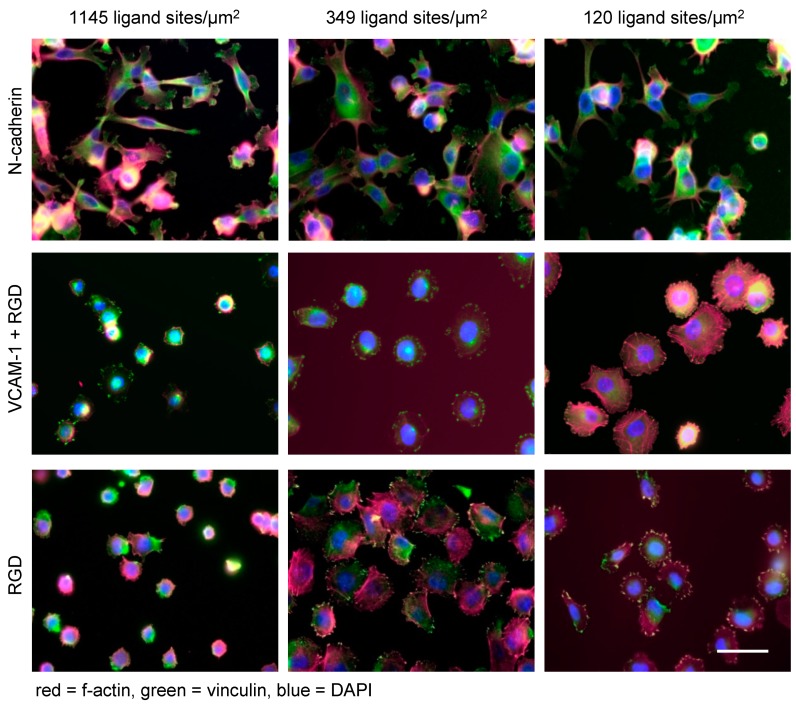
Nanotechnology unravels distinct linear and non-linear effects of extracellular ligands on melanoma cell functions. Human melanoma cells (cell line A375) spread on N-cadherin (upper panel), VCAM-1 (middle panel) and RGD (lower panel) displayed at densities of 1145, 349, and 120 ligand sites/µm^2^ (corresponding to lateral distances of 30, 60, and 90 nm), respectively (scale bar = 50 µm). The matrices were produced by identical BCML procedures. The pictures indicate that ligand density and ligand specificity are both relevant in determining the cellular fate: while melanoma cells show similar spreading on different densities of N-cadherin, VCAM-1 exerts an inhibitory effect in a density-dependent linear fashion. Finally, the integrin peptide RGD induces maximum spreading on intermediate densities of RGD peptides. These ligand-specific biophysical traits were revealed by nanotechnology and were not discernable from “conventional” ligand-coating experiments.

**Table 1 nanomaterials-10-00212-t001:** Overview of selected nanomodel-systems alluded to in the text. The table highlights the methods used, the cell lines and ligands/receptors and the major findings regarding tumor progression-related functions.

Method	Cell Type	Ligand Type	Nanoscopic Control	Tumor Progression-Related Outcome	Reference
**BCML (Block-Copolymer Nano-Lithography)**	Human melanoma cells	RGD	- Monovalent - global density - spatial distance	Definition of optimum ligand density	[77]
	Human melanoma cells	VCAM-1 (plus RGD)	- monovalent (plus RGD background) - global density - spatial distance	Antagonistic function of VCAM-1 and RGD upon cell spreading	[78]
	Human melanoma cells	N-cadherinE-cadherin	- monovalent - global density - spatial distance	Flexible spreading irrespective of density	[76]
**EBL (Electron Beam Lithography)**	Human breast cancer cells	Fibronectin (plus Laminin/K-casein background)	- monovalent - gradient	Invadipodia formation	[62]
	Human breast cancer cells	Fibronectin	- monovalent - gradient	Flexible spreading	[63]
**DNA-Origami**	Human melanoma cells	- EGF - A20FMDV2 peptide (integrin ligand)	- bivalent - ligand distance - ligand ratio	Cooperative signaling upon spreading	[55]
	Human breast cancer cells	EGF	- monovalent - ligand number and distribution	EGF ligand architecture determines cellular response	[73]

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
