# Peer review of "Modulating Tumor Cell Functions by Tunable Nanopatterned Ligand Presentation"

_nanomaterials, 2020, doi:10.3390/nano10020212_

Round 1
Reviewer 1 Report
In this review, Amschler and Schön discuss the application of nanofabrication methods to generate patterned ligand models to study tumor cell mechanobiology. The scope of the manuscript is appropriate, with consideration given to three specific techniques: block copolymer micelle nanolithography, e-beam lithography, and DNA origami. However, the submission could be improved in both clarity and utility to the reader with a number of modifications.
Major Issues
1) Figure 1 needs to be refined. Some features, including the melanoma cell nucleus and cytoskeleton, are not obviously defined. Some components of the figure (e.g. YAP/TAZ) are not discussed in the text until later, making their introduction in the figure confusing. The figure caption discusses extravasation of melanoma cells, but only contains an unlabeled inward-pointing arrow within the blood vessel. It’s unclear if the figure is meant to show tumor metastasis from a the primary site or some other process. And it is unclear what message the stiffness/topography/ligand density gradients shown at the bottom are meant to convey. Perhaps breaking up the figure to focus separately on the specific phenomena to be highlighted would be more clear.
2) The authors mention many times how ECM properties can promote or inhibit tumor cell activity without providing specific, concrete, and referenced examples. Referencing specific examples, especially in the introduction sections, would strengthen the justification of the manuscript.
3) Figure 2 needs labels for panels A-D so the reader does not need to refer to the caption to understand what is being illustrated. It was confusing for me that the fabrication methods were shown from bottom to top.
4) Addition of a table presenting how the three fabrication techniques have been used to study tumor cell behavior would help the reader understand how widely used each technique is relative to each other, and for which tumor types they have been applied. The table could include the fabrication method, tumor type, ligand type, and the pertinent findings for each study. This information would provide much-needed context to the reader.
5) The authors should also include a discussion of the limitations of current methods and models (e.g. limited to two dimensions, difficulties of co-culture with other cell types, etc.
Minor Issues
Line 162 - When the abbreviation BCML is introduced, it would be informative to include the word ‘micelle’ so it is clear from where the abbreviation is derived.
Line 171 - The equation here is introduced without defining the variables
Line 233 - loose should be lose
Author Response
Dear editors, dear Dr. Lauritano:
Please find enclosed our revised manuscript, in which we have responded to all of the reviewers’ criticisms and suggestions. We thank both reviewers for their thorough review and their valuable suggestions. According to their points, we have added or modified several text passages, cited additional references, amended two figures and added a table. In particular, the following alterations have been made (highlighted in the text in red):
In response to reviewer #1:
Q1: Figure 1 needs to be refined. Some features, including the melanoma cell nucleus and cytoskeleton, are not obviously defined. Some components of the figure (e.g. YAP/TAZ) are not discussed in the text until later, making their introduction in the figure confusing. The figure caption discusses extravasation of melanoma cells, but only contains an unlabeled inward-pointing arrow within the blood vessel. It’s unclear if the figure is meant to show tumor metastasis from a the primary site or some other process. And it is unclear what message the stiffness/topography/ligand density gradients shown at the bottom are meant to convey. Perhaps breaking up the figure to focus separately on the specific phenomena to be highlighted would be more clear.
R1: We substantially re-worked Figure 1 (p. 3) with the following specific modifications:
a) As some of the steps of tumor progression were unclear (like the process of extravasation) we highlighted separately essential steps of melanoma progression (invasion, migration, extravasation, survival in circulation and extravasation) in order to give an overview of how melanoma cells get in contact with different tissue and cells with respective contextual ligand qualities. b) The reviewer criticized the depiction of YAP/TAZ together with the presentation of the cytoskeleton. We agree with this reviewer that this approach depicted too much information and was therefore confusing. In the new figure, we deleted all reference to YAP/TAZ which was not mentioned in the text so far. c) We highlighted three different steps of cellular interactions (a, b, c) in different stages of melanoma progression. These cells are exposed to different components of the ECM regarding ligands, ligand density, stiffness and topography. Reviewer 1 criticized that it was unclear what message the stiffness/topography/ligand density gradients should tell. The aim of showing these gradients was to show that variation of these ECM properties would influence distinct functions of tumor cells. In the following some of the modifications by ECM properties (according to our data) were selected to show how they might influence spreading and the cytoskeleton. In Figure 1A melanoma cells get in contact with an ECM of high densities of RGD resulting in inhibition of cellular spreading. In Figure 1B melanoma cells show maximum spreading in an ECM containing reduced RGD ligands. In Fgure 1C melanoma cells beginning to extravasate are small and their cytoskeleton is inhibited as detected on high densities of VCAM-1. These aspects have now been explained better in the legend to figure 1.
Q2: The authors mention many times how ECM properties can promote or inhibit tumor cell activity without providing specific, concrete, and referenced examples. Referencing specific examples, especially in the introduction sections, would strengthen the justification of the manuscript.
R2: To address this point, we added a section in the introduction focusing on recent examples of how ECM properties promote tumor cell activity (p. 2, 2nd paragraph, lines 56-65). Furthermore, we have cited four additional relevant publications. This new passage together with the new references now gives a more comprehensive view that strengthens the justification of our manuscript.
Q3: Figure 2 needs labels for panels A-D so the reader does not need to refer to the caption to understand what is being illustrated. It was confusing for me that the fabrication methods were shown from bottom to top.
R3: Following the suggestion of this reviewer, we added labels for panels A-D to improve the readability (p. 6). We understand the point this reviewer made regarding our presentation of the nanomodels “from bottom to top”. However, we believe that this type of presentation will better highlight the focus of our manuscript on the structures ultimately used for biological experiments if the final products can be compared in a row at the top of the figure. We understand that this is a matter of opinion and we hope that the reviewer can accept our point of view.
Q4: Addition of a table presenting how the three fabrication techniques have been used to study tumor cell behavior would help the reader understand how widely used each technique is relative to each other, and for which tumor types they have been applied. The table could include the fabrication method, tumor type, ligand type, and the pertinent findings for each study. This information would provide much-needed context to the reader.
R4: As suggested by this reviewer we included a table presenting how the three fabrication techniques have been used to study tumor cell behavior (p. 8). The table gives an overview on what has been found out regarding tumor progression-related functions using BCML, EBL and DNA origami. In addition, we have cited four more relevant publications in this context (highlighted in the reference section).
Q5: The authors should also include a discussion of the limitations of current methods and models (e.g. limited to two dimensions, difficulties of co-culture with other cell types, etc.
R5: We absolutely agree with this reviewer and we included a discussion of limitations of the current models (p. 9, last paragraph). We also added two relevant references.
Q6: Line 162 - When the abbreviation BCML is introduced, it would be informative to include the word ‘micelle’ so it is clear from where the abbreviation is derived.
R6: The clarification has been added on first mention of the abbreviation (p. 5, 5th paragraph, line 185).
Q7: Line 171 - The equation here is introduced without defining the variables
R7: For the sake of clarity and improved readability, we have deleted the formula (p. 5, 5th paragraph, line 193).
Q8: Line 233 - loose should be lose
R8: Loose” has been corrected to “lose” (p. 7, 3rd paragraph, line 254).
In response to reviewer #2:
Q1: The manuscript by Amschler and Schon extensively review the impact of nanotechnology on evaluating membrane receptor density, role, and in general biophysics of tumor cells. The manuscript is extremely well written and pleasant to read, I could hardly find any suggestion, thus I think it could nearly be published as is. My only, minor, suggestion is related to the possible insertion of a paragraph dealing with optical nanoscopy in vitro. While this topic is touched in some part of the manuscript, it could benefit from a larger discussion, given the increasingly recognized role of this kind of approach in evaluating receptor dynamic and interplay adt nanoscale.
R1: We thank this reviewer for his/her very positive evaluation! One suggestion was to insert a paragraph dealing with optical nanoscopy in vitro. We now included a brief section on optical nanoscopy when adding a new section with the topic “limitations of the nanomodels”: Optical nanoscopy should be used to improve data evaluation when analyzing cells on nanopatterns in order to gain profound information about subcellular structures (see section “Limitations of current nanomodel-systems”, p. 9, last paragraph, lines 322-324). We also added two new references on optical nanoscopy.
We hope that you will find our revised manuscript suitable for publication in Nanomaterials.
Sincerely,
Michael P. Schön

Reviewer 2 Report
The manuscript by Amschler and Schon extensively review the impact of nanotechnology on evaluating membrane receptor density, role, and in general biophysics of tumor cells. The manuscript is extremely well written and pleasant to read, I could hardly find any suggestion, thus I think it could nearly be published as is. My only, minor, suggestion is related to the possible insertion of a paragraph dealing with optical nanoscopy in vitro. While this topic is touched in some part of the manuscript, it could benefit from a larger discussion, given the increasingly recognized role of this kind of approach in evaluating receptor dynamic and interplay adt nanoscale.
Author Response

(The authors gave the same response as above.)

Round 2
Reviewer 1 Report
I appreciate the authors' efforts to improve the manuscript. This revised version is suitable for publication. I have two final suggestions:
1) It would help if the specific references included in Table 1 were inserted into the table itself. This way the reader knows which references are associated with each nanofabrication method.
2) In many instances, the authors refer to the modulation of tumor cell spreading by changes in either native or engineered ligand density. If there is a known link between this type of cell spreading/morphology and a functional phenotype (e.g. metastatic potential, susceptibility to apoptosis, etc.), it would be worth highlighting and clarifying.
Author Response
Dear editors, dear Dr. Lauritano:
Please find enclosed our revised manuscript, in which we have responded to the additional suggestions raised by reviewer #1. The following alterations have been made (highlighted in the text in red):
In response to reviewer #1:
Q1: It would help if the specific references included in Table 1 were inserted into the table itself. This way the reader knows which references are associated with each nanofabrication method.
R1: In principle, of course, you are right. The assignment would be more unambiguous if the references were named directly in the table. We had already thought so ourselves. The thing is, however, that in this journal the references are listed numbered in the order in which they are called up in the manuscript. Since we do not know where exactly our table will be printed in the end, the numbering of the references could still shift. In order to avoid confusion, we had decided to mention them only in the text.
However, we have now also included the references in the table and ask the publisher to ensure that they are published in the correct order.
Q2: In many instances, the authors refer to the modulation of tumor cell spreading by changes in either native or engineered ligand density. If there is a known link between this type of cell spreading/morphology and a functional phenotype (e.g. metastatic potential, susceptibility to apoptosis, etc.), it would be worth highlighting and clarifying.
R2: Unfortunately, there are hardly any published data that clearly link the functions investigated in vitro with actual progression, invasion or other direct pathological functions in vivo. Many of these relationships are still hypothetical, although very probable and plausible. This is primarily due to the fact that biophysical influencing factors such as the in situ nanoscale ligand density in the in vivosituation can hardly be determined exactly. We have emphasized this difficulty in our introduction (p. 2, paragraphs 2 and 3). However, in our own recent work on the role of VCAM-1 in melanoma cells we have shown very suggestive circumstantial evidence for a significance of the in vitro spreading investigated with nanostructures with the actual tumor cell behavior in a preclinical in vivomodel. We have therefore now included a sentence emphasizing this connection even more clearly (p. 8, 1st paragraph).
We hope that you will find our manuscript now suitable for publication in Nanomaterials.
Sincerely,
Michael P. Schön
